# Design and Characterization of pMyc/pMax Peptide-Coupled Gold Nanosystems for Targeting Myc in Prostate Cancer Cell Lines

**DOI:** 10.3390/nano13202802

**Published:** 2023-10-21

**Authors:** Samuel Longoria-García, Celia N. Sánchez-Domínguez, Margarita Sánchez-Domínguez, Jesús R. Delgado-Balderas, José F. Islas-Cisneros, Oscar Vidal-Gutiérrez, Hugo L. Gallardo-Blanco

**Affiliations:** 1Departamento de Bioquímica y Medicina Molecular, Facultad de Medicina, Universidad Autónoma de Nuevo León, Monterrey 64460, Mexico; 2Centro de Investigación en Materiales Avanzados, S.C. (CIMAV, S.C.), Unidad Monterrey, Apodaca 66628, Mexico; 3Facultad de Ciencias Químicas, Universidad Autónoma de Nuevo León, Avenida Universidad s/n, Cd. Universitaria, San Nicolás de los Garza 66455, Mexico; 4Servicio de Oncología, Centro Universitario Contra el Cáncer (CUCC), Hospital Universitario “Dr. José Eleuterio González”, Universidad Autónoma de Nuevo León, Monterrey 66451, Mexico

**Keywords:** Myc, Max, cancer, nanomedicine, AuNPs, peptides, nanocomplexes, colloidal nanoparticle clusters

## Abstract

Myc and Max are essential proteins in the development of prostate cancer. They act by dimerizing and binding to E-box sequences. Disrupting the Myc:Max heterodimer interaction or its binding to E-box sequences to interrupt gene transcription represent promising strategies for treating cancer. We designed novel pMyc and pMax peptides from reference sequences, and we evaluated their ability to bind specifically to E-box sequences using an electrophoretic mobility shift assay (EMSA). Then, we assembled nanosystems (NSs) by coupling pMyc and pMax peptides to AuNPs, and determined peptide conjugation using UV-Vis spectroscopy. After that, we characterized the NS to obtain the nanoparticle’s size, hydrodynamic diameter, and zeta potential. Finally, we evaluated hemocompatibility and cytotoxic effects in three different prostate adenocarcinoma cell lines (LNCaP, PC-3, and DU145) and a non-cancerous cell line (Vero CCL-81). EMSA results suggests peptide–nucleic acid interactions between the pMyc:pMax dimer and the E-box. The hemolysis test showed little hemolytic activity for the NS at the concentrations (5, 0.5, and 0.05 ng/µL) we evaluated. Cell viability assays showed NS cytotoxicity. Overall, results suggest that the NS with pMyc and pMax peptides might be suitable for further research regarding Myc-driven prostate adenocarcinomas.

## 1. Introduction

The Myc transcription factor (TF) and its associated Max protein play critical roles in the development of various types of cancers, including prostate cancer (PCa) [1,2,3]. Myc overexpression in PCa significantly drives tumorigenesis and progression [4]. PCa is the most common cancer among men; over 1,414,000 estimated new cases were reported in 2020 [5,6]. The Myc gene encodes for the Myc TF that regulates essential cellular processes, such as nucleotide and fatty acids synthesis, glycolysis, glutaminolysis, and mitochondrial and ribosomal biogenesis [7]. Myc is classified in the family of TFs that contain a basic helix-loop-helix (bHLH) domain and a leucine zipper (LZ) domain; these domains allow the TF to interact with other TFs and proteins through dimerization. This extensive interaction network is known as the Myc interactome [8]. One of the most relevant proteins involved in the Myc interactome is the Max (Myc-associated factor X) protein.

The Myc:Max dimer recognizes a specific promoter DNA sequence in its regulated genes known as the E-box sequence. The Myc:Max dimer has a higher affinity to the E-box canonical sequence CACGTG (CME), whereas the dimer shows reduced binding affinity to degenerated sequences derived from the CME (non-CME). During the cell’s transformation into a cancerous phenotype, overexpressed Myc forms dimers with Max, and they tend to bind to non-CME and other non-E-box elements, a phenomenon known as promoter invasion [9]. Disruption of Myc:Max heterodimerization or their binding to the CME and non-CME is a current therapeutic strategy to treat cancer [10]. Currently, Omomyc, a synthetic peptide derived from Myc, is being used as a therapeutic approach to disrupt the transcription of target genes in cancer [1]. We considered the precedent strategies and proposed that an optimized peptide design using computational biology tools could target Myc and other TFs in cancer and other diseases.

However, one limitation of Myc-derived peptides is the difficulty of delivery to the nucleus. Apparently, Myc-derived peptides could have low efficiency in crossing the cellular and nuclear membranes; using peptide-functionalized nanoparticles is a strategy that can resolve this problem [11,12,13] based on three primary characteristics: (a) the size of the nanoparticles (NPs) that can be used to be delivered into the cells, (b) NPs can be functionalized with diverse compounds for theranostic purposes [14,15], and (c) NPs functionalized with Myc-derived peptides can improve molecular interactions between peptides and cell and nuclear membranes, increasing the chances of cellular import of peptide-nanoparticles compared to peptides without conjugated NPs [13,16]. Gold nanoparticles (AuNPs) have several distinctive properties that allow them to be used for biomedical applications: straightforward synthesis, strong binding affinity to thiol functional groups, surface plasmon resonance (SPR) phenomenon, and generally low toxicity [17,18,19]. These characteristics of AuNPs are very useful for drug delivery or bioimaging; furthermore, peptide-AuNPs composites (nanocomplexes) also have the possibility to be used for photodynamic or photothermal therapies [20,21]. 

Peptide-NPs conjugates have been the object of study for several years; some of their applications involve inhibition of pathogen molecular interactions with the host, drug delivery, and molecular imaging [22]. In biomarker detection, AuNPs-peptide conjugates have been used for detecting circulating tumor cells in human peripheral blood using an epidermal growth factor (EGF) peptide [23]; in bioimaging, they have been used to target β-amyloid fibers in Sprague-Dawley rats using an amphipathic peptide (aminoacidic sequence: CLPFFD) [24]. Other peptide-AuNPs conjugates have been used to deliver siRNAs to nuclei of MCF-7 cells using a nuclear localization signal (NLS) peptide [25] and plasmid DNA to melanoma tissue using the TAT cell-penetrating peptide [26], showing enhanced therapeutic results. This evidence proves the suitability of AuNPs conjugates for therapeutic purposes.

The nanocomplexes can interact as colloidal nanoparticle clusters, and each nanocomplex can be considered an “individual particle” with controlled sizes, shapes, composition, and biochemical properties. The combination of the individual properties can create new collective properties, for example by steric stabilization [27]. The objective of this work was to assemble and characterize nanosystems (NSs) composed of (AuNPs) coupled with the designed synthetic peptides derived from Myc (pMyc) and Max (pMax) to evaluate hemocompatibility and to estimate cytotoxicity in PCa adenocarcinoma cell lines.

## 2. Materials and Methods

We used the native Myc (NP_002458.2) and Max (NP_660087.1) reference sequences to design the peptides. We also used computational biology tools: Swiss-Model for protein homology modeling [28], Cn3D for rendering the macromolecular structure and residue interactions [29], and ScooP for predicting thermal stability [30]. A general overview of the editing process can be seen in Figure 1. The pMyc and pMax peptides were chemically synthesized employing Accura’s Custom Peptide Synthesis service (Accura, Monterrey, Nuevo Leon, Mexico).

To conduct the electrophoretic mobility shift assay (EMSA) and to avoid undesired oligonucleotides mismatches, we designed and synthesized ssDNA oligonucleotides using IDT’s OligoAnalyzer tool (Version 3.1) and IDT’s Custom DNA oligos service (IDT, Coralville, IA, USA). We show the oligonucleotide sequences in Table 1. The oligonucleotides have the canonical E-box sequence (CME) or non-E-Box element (NE) to which Myc has been shown to bind; we also used oligonucleotides previously used by Allevato et al. (CME Allevato) [9] and a negative control of oligonucleotides that do not contain an E-box (CTRL).

The oligonucleotides were diluted in TE pH 7.5 buffer to obtain a 5 µM final concentration. Then, they were mixed with their complementary oligonucleotides and hybridized using the following steps: 95 °C 5 min, 62 °C 20 min, and 25 °C 20 min in a standard final point MJ-mini Bio-rad thermocycler (Bio-rad, Hercules, CA, USA).

Three different EMSA gels were run combining the pMax and pMyc peptides with four different dsDNA oligonucleotides: (1) pMax, (2) pMyc, and (3) pMyc:pMax heterodimer. The EMSAs were carried out following Ream’s protocol [31] with some changes. First, a 9 × 11 cm 2.5% agarose was run at 160 V for 10 min and then at 100 V for 90 min in TB 0.5 X buffer. We captured pictures of the gels using a Chemi Camera EC3 UVP BioImaging System (Bio-rad, Hercules, CA, US). 

Then, three different NSs were assembled as illustrated in Figure 2: pMyc:AuNPs, pMax:AuNPs, and pMyc:pMax:AuNPs. Maleimide 5 nm Gold Nanoparticle Conjugation kits (cat. No 900458-1EA Cytodiagnostics, Burlington, ON, Canada) were used for assembling the NS. These nanoparticles are already functionalized with PEG-maleimide to enable conjugation and grant stability. pMyc and pMax were diluted to a working concentration of 1 mg/mL in 50 µL of 1X PBS. These were then reduced by adding 1 µL 0.5 M dithiothreitol (DTT) and incubated for 2 h in the dark at room temperature. We removed the DTT using Spin-X UF columns with a molecular weight cutoff of 5 kDa, and samples were washed twice with 1X PBS (cat. No CLS431477-25EA, Corning LifeSciences, Tewksbury, MA, USA). 

After the peptides were reduced, the conjugation was conducted according to the manufacturer´s instructions. We added the kit’s resuspension buffer. Afterward, 48 µL of each diluted peptide was mixed with 60 µL of reaction buffer, and 90 µL of this solution was transferred to the vial containing the AuNPs. Then, the vial was stored in the dark at room temperature for 1 h. After the 1 h incubation time, the reaction was stopped by adding 10 µL of the quencher solution and incubated for 15 min. The peptide-conjugated AuNPs NS were purified through centrifugation in a 100 kDa MWC Spin-X UF Corning columns to separate and eliminate unconjugated elements.

We obtained the UV-Vis spectra with an ND-1000 Nanodrop (Thermo Fisher Scientific, Waltham, MD, USA) in the wavelength range of 220 to 748 nm. In addition, we also used unconjugated AuNPs as a control.

The zeta potential was assessed based on electrophoretic mobility, and size distribution was measured using dynamic light scattering (DLS). Both properties were determined using a Zetasizer Nano ZS90 (Malvern Instruments, Malvern, UK), and dilute dispersions in water were prepared to carry out the measurements. TEM images were obtained on a 200 keV JEOL JEM2200 FS electron microscope (JEOL Ltd., Tokyo, Japan). The samples for TEM characterization were prepared by carefully placing a drop of AuNPs or NS (2.5 ng/µL) dispersion on a formvar/carbon-coated copper grid and dried at room temperature.

For the hemolysis assays, we collected 2 mL of blood from a healthy donor, followed by centrifugation at 3000 rpm for 3 min. Samples were further washed with 1X PBS thrice, discarding the supernatant without disturbing the erythrocyte layer. Afterward, a 1:99 erythrocyte solution was prepared by mixing 9.8 mL of 1X PBS and 100 µL of the previously washed erythrocytes. Three different concentrations of the NS were evaluated: 5 ng/µL, 0.5 ng/µL, and 0.05 ng/µL. Briefly, 16 µL of each NS was mixed with 4 µL of the 1:99 erythrocyte solution, and the samples were incubated for 30 min at 37 °C and centrifuged at 300 rpm in an AG 5350 Thermomixer (Eppendorf, Hamburg, Germany). After 30 min, the mixture of erythrocytes and NS was centrifuged at 14,000 rpm for an additional 3 min. The absorbance was determined with a ND-1000 Nanodrop at 415 nm. Here, 1X PBS was used as a non-hemolysis control, and distilled water as a positive hemolysis control. The hemolysis percentage was determined with the following formula:$$Hemolysis\;(\%)=\left(\frac{\left[{sample}_{Abs415}-{negative\;control}_{Abs415}\right]}{\left[{sample}_{Abs415}-{negative\;control}_{Abs415}\right]}\right)*100$$

All cells were acquired from the American Type Culture Collection (ATCC, Manassas, VA, USA). PC-3 and Vero CCL-81 cells were cultured in DMEM (Thermofisher Scientific, 11965092), and the LNCaP clone FGC and DU145 cells were cultured in RPMI 1640 (Gibco, 72400047). LNCaP clone FGC, PC-3, and DU145 cells were chosen since they were obtained from a prostatic adenocarcinoma lineage. The Vero CCL-81 cell line was used to measure the cytotoxic effect in a non-cancerous cell line. RNA-Seq and proteomics data on Myc mRNA and relative protein expression in cell lines were extracted from DepMap Portal, as seen in Table 2 (https://depmap.org/portal/interactive/ (accessed on 20 February 2023).

Cell viability was measured using the Cell Proliferation Kit I (Roche, 11465007001). First, in a 96-well plate, 10,000 cells were seeded and incubated for 24 h (Vero CCL-81, PC-3, and DU145) or 48 h (LNCaP), depending on the cell line. Next, cells were treated with 5 ng/µL, 0.5 ng/µL, or 0.05 ng/µL of the AuNPs or NS for 24 h. After 24 h, the media was replaced with new media containing 10 µL of MTT per 100 µL of media and incubated at 37 °C for 3 h. Finally, the formazan crystals were solubilized with isopropanol (pH = 3) and quantified in a BioTek Cytation3 Imaging reader at 570 nm and 651 nm.

Two-way ANOVA with the Bonferroni correction was utilized for this work’s statistical analysis. We used GraphPad Prism 9 for statistical analysis and graph creation (GraphPad Software, LLC. Boston, MA, USA).

The study was conducted according to the guidelines of the Declaration of Helsinki and approved by the Ethics Committee of the School of Medicine of the Universidad Autónoma de Nuevo León (protocol code BI22-00002 with an approval date of 11 April 2022).

## 3. Results and Discussion

### 3.1. In Silico Peptide Design and Chemical Synthesis

The designed pMyc resulted in a 51-amino acid sequence; it contains the bHLH domain for E-box sequence recognition and dimerization properties, an NLS to promote nucleus import, and a cysteine in the C-terminal end for conjugation to AuNPs. The designed pMax resulted in a 46-amino acid sequence containing the bHLH domain and a cysteine C-terminal for conjugation to AuNPs. Using SWISS-MODEL (https://swissmodel.expasy.org/interactive (5 August 2020), we performed predictions of the protein models of pMyc and pMax structures and heterodimerization [28]. The Cn3D rendering is depicted in Appendix A. The predicted pMyc:pMax heterodimer maintains the site of recognition and binding to the CME; the C-terminals of pMyc and pMax are separated by 15.3 Angstroms. ScooP webserver algorithm 1.0 predicted thermal stability of individual peptides and the pMyc:pMax dimer (Table 3) at temperatures over the standard physiological temperature of 37 °C. These results suggest that the pMyc and pMax peptides and their heterodimers would remain stable during the incubation periods in cell viability assays.

### 3.2. Peptide–Nucleic Acid Interactions via Electrophoretic Mobility Shift Assay

Notably, the homodimers pMyc:pMyc and pMax:pMax do not interact with the CME; we propose that pMyc and pMax bind to DNA only as heterodimers (pMyc:pMax) in the presence of CME sequences (Figure 3A,B). EMSAs revealed compelling peptide–nucleic acid interactions between pMyc:pMax heterodimers and CME, indicating specific binding. This can be seen in Figure 3C, specifically in lanes 4 and 8, with our designed CME oligonucleotides and CME oligonucleotides reported by Allevato and colleagues in 2017 [9].

There is also no evidence for interactions of the non-CME sequences with pMyc:pMyc, pMax:pMax, and pMyc:pMax dimers in the EMSA results (Figure 3). These results suggest heterodimer formation, a specific recognition condition, and an interaction between the pMyc:pMax dimers and the CME sequence. These findings resemble Omomyc’s ability to bind specifically to CMEs [34,35] and the results of the synthesized Myc peptide analog [AQ] Myc [36]. Our results showed a relatively low shifted distance between CMEs with the pMyc:pMax dimers concerning other sequences. The discrete shift change is primarily due to the size of the designed peptides (<7 kDa for each peptide). In contrast, in EMSAs with antibodies (~150 kDa, and 10 nm in size) or larger proteins, the shift is much more significant than the shift observed in Figure 3C [37,38].

### 3.3. UV-Vis Analysis of the Nanosystem Assembly

Conjugating the peptides to AuNPs resulted in the shift of the absorption maximum to a longer wavelength than AuNPs (Figure 4). The wavelength increased by 6 nm for pMyc:pMax:AuNPs and 9 nm for pMyc:AuNPs and pMax:AuNPs. UV-Vis analysis revealed bathochromic shifts, which indicate a change in the SPR of AuNPs, and it means that the electronic properties of AuNPs are affected by ligands [39], in this case the pMyc and pMax peptides. These bathochromic shifts have been observed in other works involving the conjugation of AuNPs with peptides (nanocomplexes) and have been proposed to indicate a successful NS assembly [40,41,42]. The results provide evidence of successful peptide and AuNPs conjugation. 

### 3.4. Nanoparticle Characterization

TEM representative images of the AuNPs and NS are shown in Figure 5, Figure 6, Figure 7 and Figure 8. Average Au core nanoparticle sizes as obtained from TEM, hydrodynamic diameters (HD, from dynamic light scattering), and zeta potential (from electrophoretic mobility) are shown in Table 4.

The TEM images (Figure 5, Figure 6, Figure 7 and Figure 8) show spherical AuNPs that ranged from 3.28 ± 1.16 nm to 5.78 ± 1.41 nm. Interestingly, the HD obtained by DLS shows diameters ranging from 225.03 ± 17.65 nm to 264.97 ± 4.39 nm. In contrast with our results, Satriano et al. reported only a slight increase in their AuNPs’ HDs from 12 nm to 60 nm. There were minimal differences between their AuNPs and the HD of their NS [43]. Similar results are presented by Taha et al., where they used 4 nm AuNPs conjugated with TAT peptide and obtained a HD of ≈ 14 nm [44].

The 5 kDa PEG-coated AuNPs are stabilized by a steric mechanism, arising from the different organic layers (polyethylene glycol chains and the peptides). The NSs are dispersed and not aggregated due to the steric effect from the organic layers, rendering the NPs colloidally stable. The zeta potential is only slightly negative in all NSs; however, even though it is relatively low (from −5 mV to −10 mV), the NSs are still colloidally stable thanks to the steric stabilization. Also, it is important to mention that the NSs were dispersed in PBS; thus, the pH of the system was approximately 7.4. It is possible that at different pH values, the zeta potential increases to more negative values; however, for biomedical applications, the NS should be characterized at physiological pH.

In 2016, Chang et al. characterized AuNPs in dispersed and assembled states with UV-Vis and TEM, and their results are in agreement with ours. The UV-Vis spectra of the NSs presented in their study showed that the SPR peak was maintained, whereas the TEM images show little to no aggregates [45]. The HD results suggest that the NSs exhibit a colloidal behavior [46,47]. 

Based on UV-Vis, DLS, and TEM image results in Figure 5, Figure 6, Figure 7 and Figure 8, the NSs were associated as colloidal nanoparticle clusters [48]. TEM image analysis focuses on the metallic core of the NS (the AuNPs), and DLS analysis reports the hydrodynamic diameter of the NS, which is formed by assemblies of AuNPs (colloidal nanoparticle clusters). The differences can be explained by the formation of nanocomplexes of self-assembled AuNPs with an organic coating formed by PEG and the corresponding peptides in the NS. Figure 5, Figure 6, Figure 7 and Figure 8 and Appendix A show TEM images and ImageJ image analysis data; it is possible to visualize the structure of formvar/carbon-coated copper grid and the colloidal nanoparticle clusters deposited on it. TEM image analysis, DLS, and UV-Vis results suggest that our AuNPs and NS are forming colloidal nanoparticle clusters [48]. Furthermore, in Appendix A we can observe that throughout the course of 96 h, there was no change in the SPR; these results suggest that the AuNPs and the NS are stable. Raw data graphs for HD and zeta potential are presented in Appendix A. 

### 3.5. Hemolysis Test

Three different NS concentrations (5, 0.5, and 0.05 ng/µL) were tested to evaluate the hemocompatibility and determine if unspecific interactions with the erythrocyte membrane might cause cytotoxicity. According to the Standard Practice for Assessment of Hemolytic Properties of Materials norm ASTM-F756-17 [49], materials are classified as follows: a hemolytic index of 0 to 2% indicates that the materials are non-hemolytic; 3 to 5% indicates slightly hemolytic materials, and 5% or greater indicates highly hemolytic materials.

Hemolysis tests demonstrated minimal hemolytic activity for the NS at the evaluated concentrations (5, 0.5, and 0.05 ng/µL), a concentration range that is between 16 nM to 224 nM peptides for all NSs. This suggests their potential intravenous biocompatibility and justifies further investigation. AuNPs at all concentrations showed the highest hemolytic results (Figure 9). The AuNPs and pMyc:pMax:AuNPs showed hemolysis effects in a dose-dependent manner. The findings indicate that combining the AuNPs with the peptides could change the NS’s interaction with the erythrocyte’s membrane. All NSs are considered non-hemolytic materials at the 0.05 ng/µL concentration. At 0.5 ng/µL, pMyc:pMax AuNPs are considered slightly hemolytic, whereas pMyc:AuNPs and pMax:AuNPs are non-hemolytic materials. Finally, at 5 ng/µL, pMyc:AuNPs and pMyc:pMax:AuNPs are considered slightly hemolytic, whereas pMax:AuNPs are considered non-hemolytic materials. 

The highest hemolytic value obtained with only AuNPs was 5 ng/µL at 5.89 ± 2.02%. The hemolytic values for the pMyc:AuNPs (4.43 ± 0.92%) and pMyc:pMax AuNPs (4.12 ± 1.22%) NS at 5 ng/µL are similar to other results reported previously with peptide-AuNPs conjugates, with hemolytic values of <10% [50]. In 2019, Verimli et al. found hemolytic values < 1% for AuNPs conjugates with an apoptotic peptide (RLLLRIGRR-NH2) at the nanomolar scale [51].

Other studies have reported hemolysis values greater than 20%. However, this might be primarily due to the range of concentrations used in the experiments, which were generally higher than ours [52,53]. Comparisons of hemolysis values are difficult because of the extensive range of materials and concentrations that need to be looked at for the tests. The newly synthesized nanostructures evaluated in this study had less than 5% hemolysis, showing that these concentrations do not substantially disrupt the erythrocyte membrane.

### 3.6. Cell Viability Assessment

pMyc:pMax dimers not coupled to AuNPs had no significant effect on all evaluated cell lines at 5, 0.5, and 0.05 ng/µL concentrations; the highest cytotoxic effect was 3% (Appendix A). Based on these results, the treatment with pMyc:pMax peptides without conjugation to AuNPs was ineffective as a cytotoxic therapy. Nevertheless, pMax:AuNPs, pMyc:AuNPs, and pMyc:pMax:AuNPs had an appreciable toxic effect on cell lines. The cytotoxicity of AuNPs, pMax:AuNPs, and pMyc:AuNPs in the Vero CCL-81 cell line was evident (Figure 10A). Cell viability was reduced to 65 ± 5.02% when 5 ng/µL of AuNPs was tested. pMax:AuNPs reduced cell viability to 77.63 ± 1.63%, and pMyc:AuNPs reduced cell viability to 63.06 ± 8.12% at the same concentration. Conversely, pMyc:pMax:AuNPs NS had a small cytotoxic effect, only decreasing cell viability to 95.89 ± 7.73% at 5 ng/µL. 

As discussed by Sani et al. in 2021, there are variable results regarding AuNPs in Vero cells, which may depend on the bioactivity of conjugated molecules [54]. Due to the bioactivity of conjugated molecules, these variable results can explain that AuNPs in certain studies showed cytotoxicity [55,56], whereas others displayed no significant cytotoxicity. Ultimately, the size and charge of AuNPs significantly influence the cytotoxic effects in Vero cells [57,58].

In our experimental design, we expected a higher sensitivity of LNCaP cells to pMyc:AuNPs and pMyc:pMax:AuNPs according to the data in Table 2 due to the higher Myc normalized reads per million (nRPM). Inhibiting these oncogenic Myc proteins should have a greater effect on cell proliferation, as seen in the results.

The cytotoxic assay results confirm our theoretical prediction; in the LNCaP cell line (Figure 10B), the most significant cytotoxic effect at 5 ng/µL was observed with pMyc:AuNPs, with a decrease in cell viability to 79.98 ± 8.00%. pMyc:pMax:AuNPs reduced cell viability to 84.41 ± 7.31%. Using several pMyc:pMax disruption compounds, Carabet et al., in 2018, reduced LNCaP cell viability to <20%. In 2022, Holmes et al. used a small-molecule Myc inhibitor (MYCi975) to reduce cell viability to 28.4%. Their assays involved treatments in the order of micromolar concentrations [59,60].

In PC-3 cells, we saw little cytotoxicity (Figure 10C). However, pMyc:pMax:AuNPs were the only material with a cytotoxic effect, decreasing cell viability to 92.9 ± 8.10% at 5 ng/µL. Interestingly, PC-3 has a higher Max nRPM (502.38) compared to Myc’s nRPM (256.91), according to the DepMap portal (Table 2). These results may suggest resistance to Myc-driven cell proliferation because of the higher amount of Max protein in these cells. 

In DU145 cells (Figure 10D), both AuNPs and pMyc:AuNPs had significant cytotoxic effects; cell viability was reduced to 84.97 ± 2.04% (AuNPs) and 82.98 ± 1.42% (pMyc:AuNPs) at 5 ng/µL. However, a significant cytotoxic effect in DU145 cells was achieved by the pMax:AuNPs, which decreased cell viability to 59.84 ± 1.38%. In 2004, Cassinelli et al. showed that the DU145 cells had a lower amount of Myc protein than PC-3 cells [61]. In 2010, Kim et al. identified that the Myc expression level in DU145 cells is on par with Myc expression levels in LNCaP cells [62]. These reports show the complexity of Myc and Max’s molecular dynamics in different cell lines. However, it is essential to note that the results presented by Kim et al. were qualitatively obtained through Western blots. In addition, DepMap portal data show low relative Myc protein expression (Table 2); this might suggest that pMax:AuNPs deplete native Myc proteins in these cells and have a cytotoxic effect in this way.

In Appendix A, it is possible to compare the highest cytotoxicity obtained with the NS described in this article and the physical characteristics of the NS. Both AuNPs and the NS have similar sizes, HDs, and zeta potentials; these results suggest that the difference in cytotoxicity is based on the different peptides that are conjugated to the AuNPs. As shown in Appendix A, the peptides on their own do not show cytotoxicity towards any of the cells used in this study; however, once they have been conjugated to AuNPs, they show different cytotoxic effects dependent on the cell line. The cytotoxicity results suggest that the NSs were internalized by one of the following mechanisms: macropinocytosis, the barrel-stave model, or the carpet-like model [13]. The macropinocytosis and carpet-like model are the most probable mechanisms for our nanocomplexes because of their high HD and given it has been proven that other large nanostructures use this endocytic pathway [63]. Based on the literature [13,63,64,65,66], the hypothesized internalization mechanism of these NS is shown in Figure 11.

## 4. Conclusions

The EMSA results showed that the designed synthetic pMyc and pMax peptides can bind to CME sequences. In addition, the UV-Vis spectra of the NS showed a bathochromic effect, which has been used as an indicator of peptide-AuNPs conjugation. Cytotoxic assays of pMyc:pMax peptides without AuNPs reveal that these materials have no cytotoxic effect. Despite this, when the NS was applied to the cancer cell lines, different response were obtained based on cell linage. We have provided evidence for the design of novel peptides and nanosystems, their specificity for E-box sequences in EMSAs, and their physical characteristics. An initial assessment of the biological effects (cytotoxicity) they exert on different prostate cancer cell linages was performed. In a cell line susceptible to bromodomain inhibitors (Vero CCL-81), bromodomain inhibition suppresses MYC gene expression. It is especially interesting that all the NSs and AuNPs possess similar physical characteristics, such as hydrodynamic diameter and zeta potential, and, yet, they still have different biological effects on different cells. This suggests that the effect they have can be attributed to the pMyc and pMax peptides, which were derived from Myc and Max proteins, as these peptides have specific structures and interactions with other molecules, proteins, and DNA E-box sequences. The differential biological effects based on cell linage were dependent in their specific molecular interactions and cell linage genetics. Each evaluated cell line has a specific genome structure and regulation as well as a specific epigenome. The pMyc:pMax:AuNPs, pMyc:AuNPs, and pMax:AuNPs were associated as colloidal nanoparticle clusters.

The changes in the interaction with the erythrocyte membrane after combining AuNPs with the peptides add essential insights. The comparison of hemolytic values to previous studies and the consideration of different materials at various concentrations provide a comprehensive analysis of the findings. All NSs are biocompatible at the concentrations tested. More information on each cell line’s Myc interactome could explain the cytotoxic effects of the different NSs. Further studies that include a broader range of concentrations, more extended treatment periods, and assessments of molecular mechanisms should be conducted to elucidate the effects of the different NSs on each cell line. 

## Figures and Tables

**Figure 1 nanomaterials-13-02802-f001:**
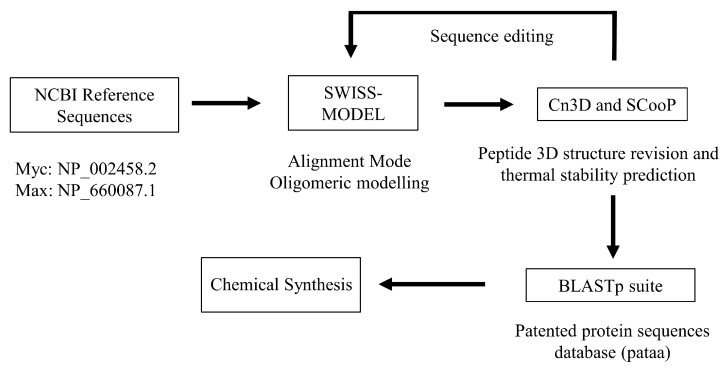
Peptide design workflow. The iterations of sequence editing were performed until the desired 3D structure and desired predicted thermal stability were obtained.

**Figure 2 nanomaterials-13-02802-f002:**
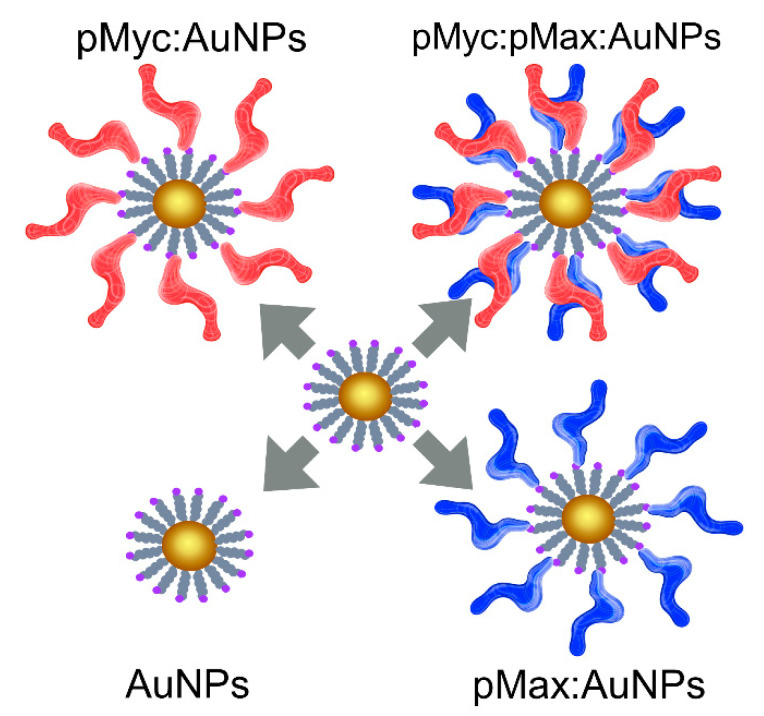
Schematic depiction of the assembled nanosystems with pMyc (red), pMax (blue), or pMyc:pMax heterodimers.

**Figure 3 nanomaterials-13-02802-f003:**
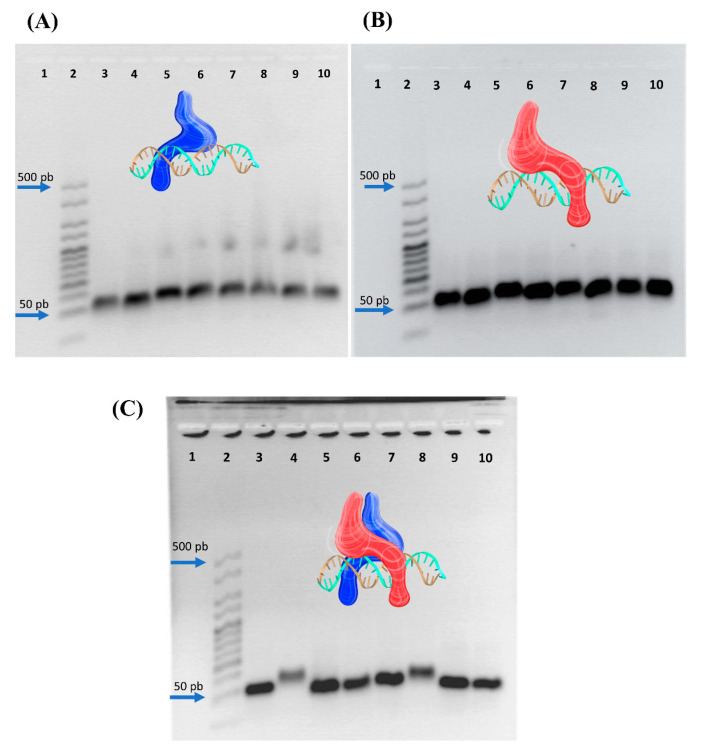
EMSAs using pMax (**A**), pMyc (**B**), and pMyc:pMax (**C**) with all of the oligonucleotide sequences described in Table 1 as follows: 1, empty; 2, Bioline Hyperladder (25 base pairs); 3, CME; 4, CME and peptide(s); 5, Ctrl; 6, Ctrl and peptide(s); 7, CME-Allevato; 8, CME-Allevato and peptide(s), 9, NE; 10, NE and peptide(s) [16]. A representation of heterodimer pMyc:pMax interacting with CME DNA was included; a representative model of pMyc in red color and blue color for a representative model of pMax, in cyan and gold color the double-strand DNA.

**Figure 4 nanomaterials-13-02802-f004:**
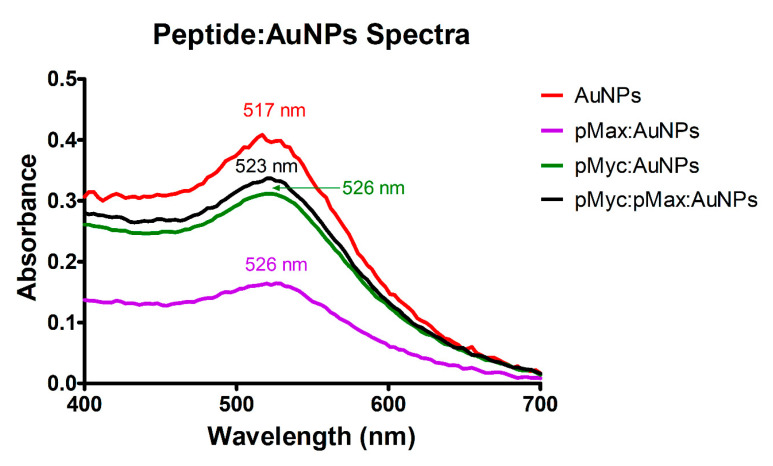
UV-Vis spectra were obtained from the AuNPs and the assembled nanosystems where the bathochromic effect can be observed.

**Figure 5 nanomaterials-13-02802-f005:**
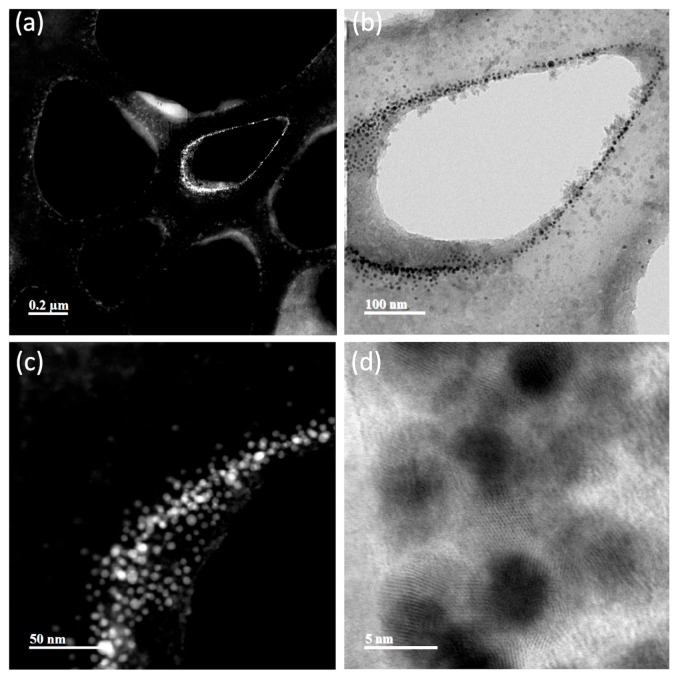
TEM images of AuNPs. (**a**) The grid structure is covered with the 5 kDa PEG-coated gold nanoparticles, forming a colloidal suspension instead of clumping together (PEG-coated AuNP particles are visible here as a pale cloud). (**b**) Colloidal structure composed of AuNPs deposited on the grid. (**c**) A magnification of the colloidal structure of AuNPs. (**d**) The 5 nm gold core appears as dark points (the same 5 nm gold core is observed as bright points in (**a**,**c**)).

**Figure 6 nanomaterials-13-02802-f006:**
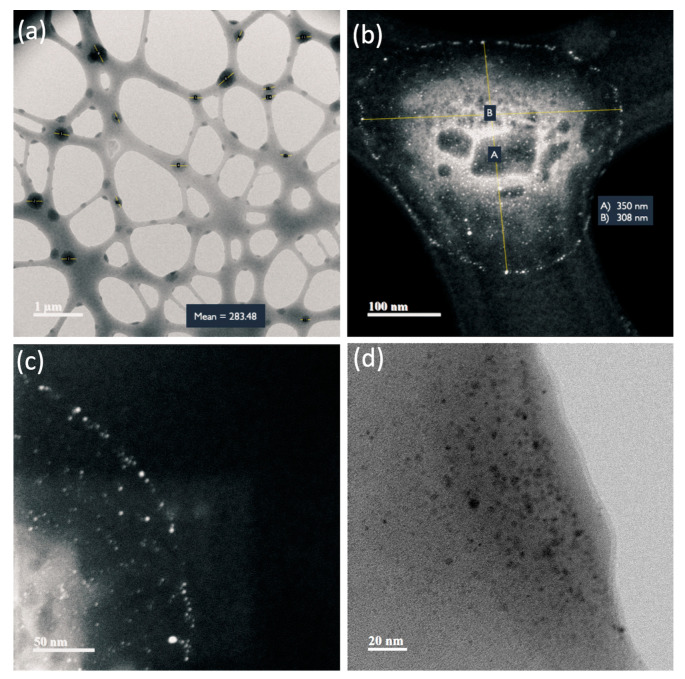
TEM images of pMyc:AuNPs. (**a**) The grid structure is covered with the pMyc-functionalized 5 kDa PEG-coated gold nanoparticles forming a colloidal suspension instead of clumping together (PEG-coated AuNP particles appear as a pale cloud). (**b**) Colloidal structure composed of pMyc:AuNPs deposited on the grid. (**c**) A magnification of the colloidal structure of pMyc:AuNPs. (**d**) The 5 nm gold core appears as dark points (the same is observed as bright points in (**b**,**c**)).

**Figure 7 nanomaterials-13-02802-f007:**
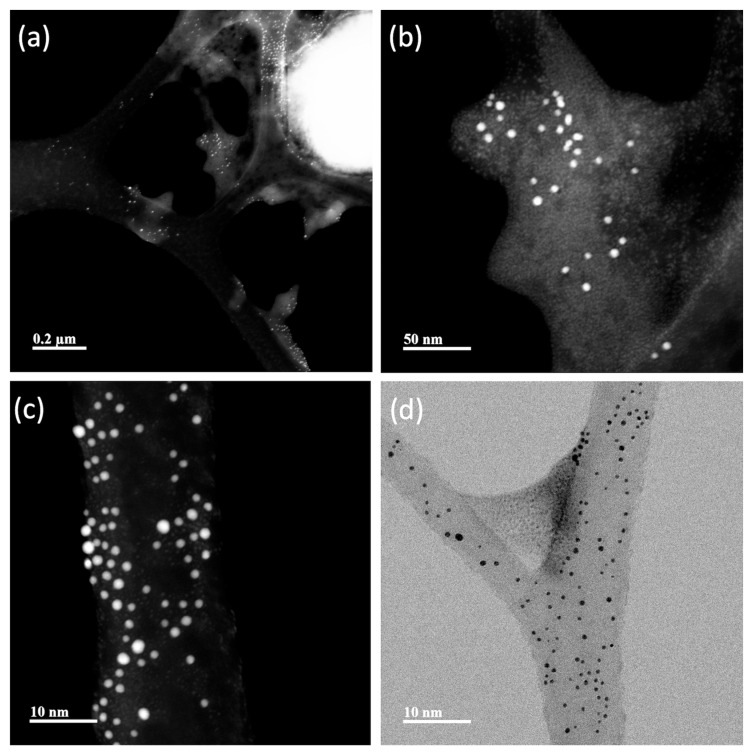
TEM images of pMax:AuNPs. (**a**) The grid structure is covered with pMax-functionalized 5 kDa PEG-coated gold nanoparticles forming a colloidal suspension instead of clumping together (PEG-coated AuNP particles appear as a pale cloud). (**b**) Colloidal structure composed of pMax:AuNPs deposited on the grid. (**c**) A magnification of the colloidal structure of pMax:AuNPs. (**d**) The 5 nm gold core appears as dark points (the 5 nm gold core appears as bright points in (**a**–**c**)).

**Figure 8 nanomaterials-13-02802-f008:**
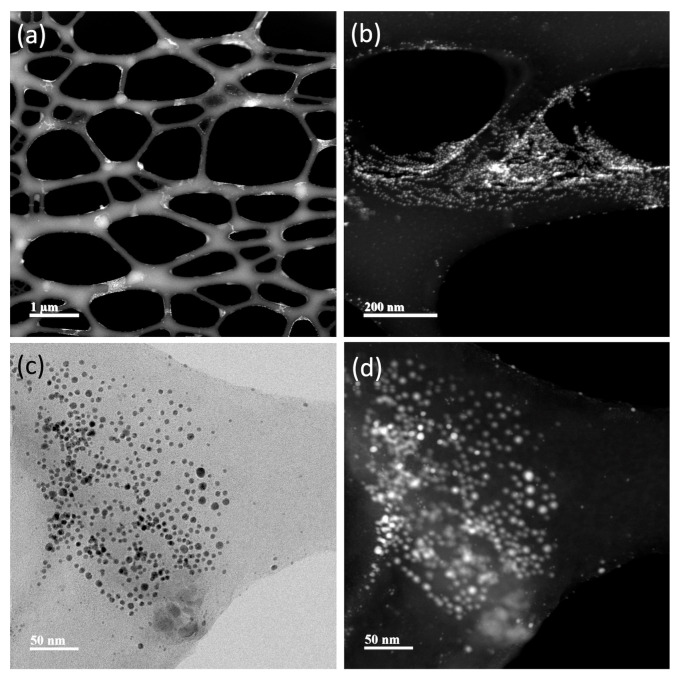
TEM images of pMyc:pMax:AuNPs. (**a**) The grid structure is covered with the pMyc- and pMax-functionalized 5 kDa PEG-coated gold nanoparticles forming a colloidal suspension instead of clumping together (PEG-coated AuNP particles are visible here as a pale cloud). (**b**) Colloidal structure composed of pMyc:AuNPs deposited on the grid. (**c**) The 5 nm gold core appears as dark points (the 5 nm gold core appears as bright points in (**a**,**b**,**d**)).

**Figure 9 nanomaterials-13-02802-f009:**
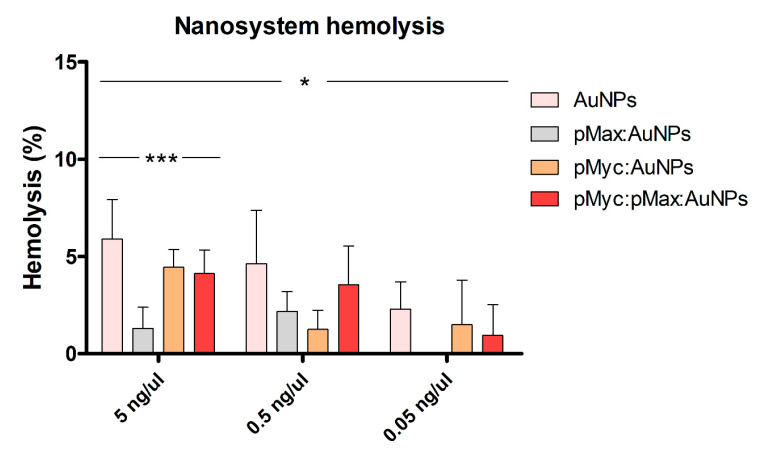
Hemolytic properties of the different nanosystems at three different concentrations. * *p* < 0.05; *** *p* < 0.001.

**Figure 10 nanomaterials-13-02802-f010:**
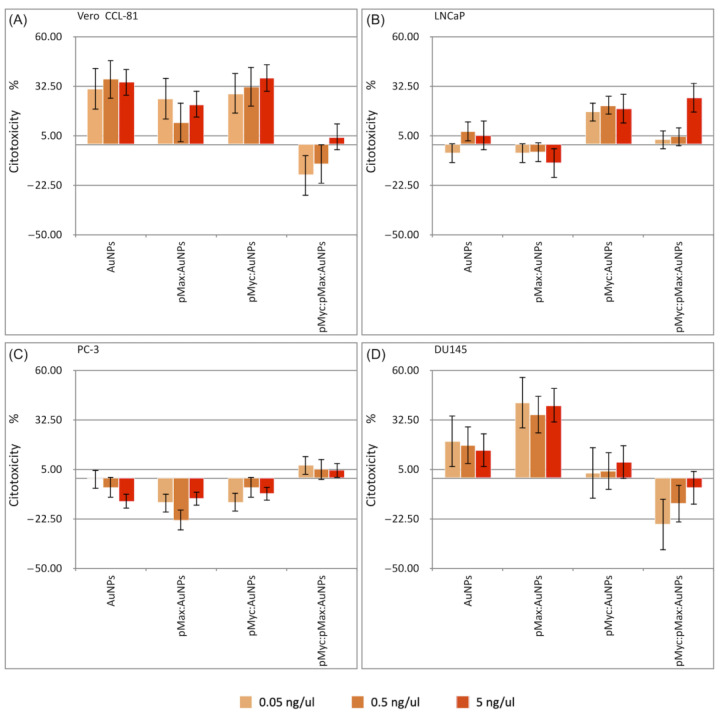
Cell viability assays with AuNPs or the different nanosystems in the four different cell lines. (**A**) Vero CCL-81, (**B**) LNCaP, (**C**) PC-3, and (**D**) DU145 cells.

**Figure 11 nanomaterials-13-02802-f011:**
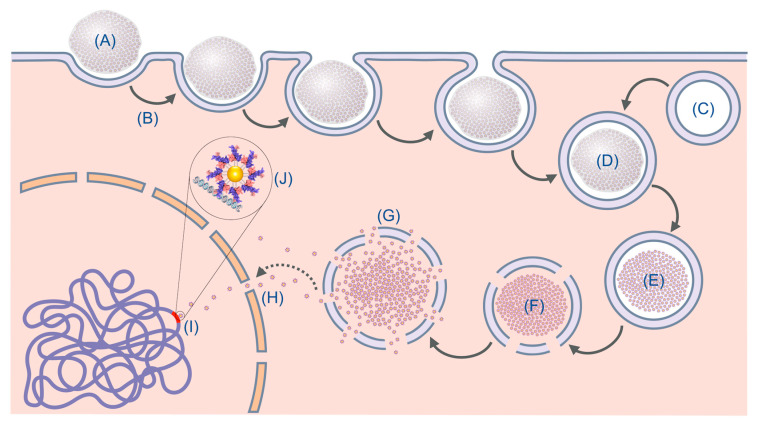
Macropinocytosis model for the internalization mechanism of our NS. (A) Nanocomplex. (B) Macropinocytosis. (C) Lysosome. (D) Early endosome. (E) Late endosome. (F) Reactive oxygen species lysosome. (G) Nanosystem liberation. (H) Nuclear transport. (I) Nanosystem interaction with DNA. (J) Close-up of the theoretical interaction of the nanosystem with DNA.

**Table 1 nanomaterials-13-02802-t001:** Designed oligonucleotides for the EMSAs. IC, oligonucleotide identification code; CME, canonical E-box; NE, non-E-box element; CTRL, control. CME sequences, target E-box sequence; CME and non-CME sequences underlined; F, forward sequence; R, reverse sequence (Adapted from [16]).

IC	Sequence
CME Allevato F	5′-CCG GCC ACG TGC ACG TGT TAA TAG CTC AGA CTA CTG TGT CGA CG-3′
CME Allevato R	5′-CGT CGA CAC AGT AGT CTG AGC TAT TAA CAC GTG CAC GTG GCC GG-3′
CME F	5′-AGA TCT CGA GCT GCA TGC TGT ACA CGT GAT GTC GTA CGT CGA GCT CTA GT-3′
CME R	5′-ACT AGA GCT CGA CGT ACG ACA TCA CGT GTA CAG CAT GCA GCT CGA GAT CT-3′
NE F	5′-AGA TCT CGA GCT GCA TGC TGT AAA CGT TAT GTC GTA CGT CGA GCT CTA GT-3′
NE R	5′-ACT AGA GCT CGA CGT ACG ACA TAA CGT TTA CAG CAT GCA GCT CGA GAT CT-3′
CTRL F	5′-AGA TCT CGA GCT GCA TGC TGT ATT AGC AAT GTC GTT ATC AGA GCT CTA GT-3′
CTRL R	5′-ACT AGA GCT CTG ATA ACG ACA TTG CTA ATA CAG CAT GCA GCT CGA GAT CT-3′

**Table 2 nanomaterials-13-02802-t002:** Myc and Max mRNA expression and relative protein expression levels in the different cell lines used in this work. nRPM, normalized reads per million; NA, not available. ^(a)^ Quantitative profiling of thousands of proteins by mass spectrometry across 375 cell lines from the Gygi lab. Proteomics [32].

Cell Line	Myc mRNA (nRPM)	Max mRNA (nRPM)	Myc Relative Protein Expression ^(a)^	Max Relative Protein Expression ^(a)^	Myc/Max Relative Protein Expression ^(a)^
PC-3	256.91	502.38	0.63	1.51	0.42
DU145	NA	NA	0.00	0.08	0.00
LNCaP Clone FGC	1548.99	315.14	0.96	0.33	2.95
Vero CCL-81	NA	NA	NA	NA	NA

**Table 3 nanomaterials-13-02802-t003:** Thermal stability prediction for the designed peptides [16].

	T_m_ (°C)	ΔG_r_ ^1^ (kcal/mol)
pMyc	80.1	−8.4
pMax	81.4	−6
pMyc:pMax dimer	76.6	−4.2
Native Myc	66.4	−5.1
Native Max	69.2	−13
Omomyc	70.4	−5.1

^1^ ScooP outputs predict the ΔGr at room temperature. The Omomyc estimate is based on the 5150.pdb structure file. Myc (Uniprot P01106) and Max (Uniprot P61244) estimates are based on the 1NKP.pdb structure file (PDB DOI https://doi.org/10.2210/pdb1NKP/pdb) (accessed on 20 February 2023) [33].

**Table 4 nanomaterials-13-02802-t004:** Nanosystems physical characterization.

	Diameter of AuNPs Core (nm)	Hydrodynamic Diameter of Nanocomplexes(nm)	PDI	Zeta Potential	Peptide Concentration (nM) ^1^
AuNPs	5.68 ± 0.84	243.03 ± 12.83	0.257 ± 0.04	−9.05 ± 6.09	0.00
pMyc:AuNPs	3.28 ± 1.16	264.97 ± 4.39	0.177 ± 0.02	−5.05 ± 3.42	163.60
pMax:AuNPs	5.60 ± 1.19	262. 57 ± 23.40	0.168 ± 0.02	−7.67 ± 5.71	200.86
pMyc:pMax:AuNPs	5.78 ± 1.41	225.03 ± 17.65	0.083 ± 0.04	−10.09 ± 4.04	223.93

^1^ Peptide concentration in the NS is presented in nM as reference. For reference, 5 ng/µL of 5 nm AuNPs is equal to 6.94 nM of AuNPs.

## Data Availability

Data are available upon request.

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
