# Peer review of "Design and Characterization of pMyc/pMax Peptide-Coupled Gold Nanosystems for Targeting Myc in Prostate Cancer Cell Lines"

_nanomaterials, 2023, doi:10.3390/nano13202802_

Round 1

Reviewer 1 Report

In this work, the authors synthesized new peptides pMyc and pMax from reference sequences and evaluated their ability to bind specifically to E-box sequences using electrophoretic mobility shift assay (EMSA). In addition, the authors obtained conjugates of their synthesized peptides with gold nanoparticles. The obtained conjugates were characterized by TEM, DLS and optical spectroscopy. In addition, the authors investigated some biological properties of the obtained conjugates, namely hemolytic compatibility and cytotoxicity towards three different prostate adenocarcinoma cell lines. In general, the work is well structured, but there are a number of comments and suggestions:

1) In the introduction, the authors should explain their choice of peptide conjugates with gold nanoparticles rather than other metals as the objects of study.

 2) In the introduction, the authors should provide an overview of the current state of research in the field of peptide conjugates with metal nanoparticles.

3) The formula on page 5 should specify the units of measurement.

4) The authors should explain the reason for the red shift of the absorption maximum in the spectra of gold nanoparticle conjugates compared to native nanoparticles.

5) The authors should explain the huge differences in the size of the nanoparticles between TEM and DLS data.

6) Low values of zeta potential of nanoparticles indicate their low aggregative and kinetic stability. How the authors solved the problem of stability of the obtained conjugates.

7) The authors should explain how optical spectroscopy indicates the absence or presence of aggregates in solution? Only DLS can indicate the presence of aggregates in solution. A presentation of the DLS data is required,

8) The authors point out the effect of charge as one of the reasons for cytotoxicity of nanoparticles and their conjugates, but according to the presented data there is no special difference between the charges of the obtained samples, and the zeta potential value is almost the same.

9) The authors should add data on the synthesis of gold nanoparticles and their characterization.

Minor correction of the text by a native English speaker is required.

Reviewer 2 Report

This work is based on the development of nanosystems based on pMyc/pMax peptide-coupled gold nanoparticles for prostate cancer treatment. However, it seemed that the characterization of NS was not sufficient and Results and discussions were not organized and presented well. It is thought that this work needs major revision to be accepted for publication. The details are as follows.

 1. The authors need to present the detailed information of pMyc and pMax for readers to understand this work well. It can also affect the purification efficiency during purification and explain the size difference between DLS and TEM analysis.

2. What is the conjugation degree of peptides onto AuNPs? It is important because it can affect the physicochemical properties and biological activity of NS such as hemolysis and cell viability.

3. What is the reason that AuNPs are accumulated in one side in Figure 5? The authors need to show clearer and more distinct TEM images.

4. In the case of pMyc:pMax:AuNPs formation, it is thought that some peptides may bind to NS by dimerization, not by chemical conjugation. It can decrease the stability and activity of this NS. How do the authors purify or characterize this?

5. For pMyc and pMax to bind to E-box sequences, NS should be internalized and release the peptides in cells. What is the mechanism for those processes? Have the authors examined these?

6. What is the reason that the peptides showed higher RCV in higher concentration in Figure S2? Also, Figure S2 lacks the detailed caption.

7. It seems that the result is not enough for evaluation of the NS in Figure 7. As the authors mentioned in Conclusions, it is required to perform further studies.

Round 2

Reviewer 1 Report

The authors have taken into account the reviewers' wishes and provided detailed responses to them. However, there are a number of wishes to the text, which allow to improve the manuscript:

1) In the paragraph of the manuscript devoted to the review of the current state of research in the development of peptide metal conjugates, the authors, in addition to listing potential applications, should have specified the types of peptide ligands used for this purpose.

2) In their response to comment 5, the authors repeatedly mention the formation of nanocomplexes between AuNPs. The authors should use more common terminology and call nanocomplexes apparently associates. 

3) In their response to comment 6 and in the text of the manuscript, the authors point out that there is no correlation between the zeta potential and the stability of nanoparticles. I absolutely cannot agree with this remark. In this case, the authors should remove the data on zeta potential measurements altogether, because the results obtained contradict the generally accepted principles of particle stabilization. 

4) I continue to insist on the predominant use of DRS for the characterization of aggregation processes, since neither TEM nor optical spectroscopy are specific methods for this purpose. In their reply and the text of the manuscript, the authors do refer to a number of papers that attempted to characterize aggregated nanoparticles by optical spectroscopy. However, for the purity of the experiment, the authors should experimentally obtain extinction coefficients for nanoparticles of different sizes in order to correctly interpret the obtained spectroscopic data. 

5) In this paper, the authors obtained zeta potential values of a number of AuNPs peptide conjugates, which practically do not differ from each other in their values, so it makes no sense to discuss the possible influence of the charge of such NS on their cytotoxicity and, moreover, to explain the difference in their activity. In this case, the authors should focus on other possible reasons for the difference in the expression of cytotoxicity of the obtained conjugates.

Reviewer 2 Report

It seems that the authors replied to several comments reasonably. However, there were still some issues to be resolved.

1. what does the concentration of each peptide mean in Table 4? Is it a concentration of each peptide used for the conjugation reaction? If so, the authors need to evaluate the exact conjugation degree of peptide onto AuNPs

2. I can’t find supplementary Table S3. There were only supplementary Figures and QC reports in Supplementary files.

3. Simple cell viability assay is not enough for evaluation of the NS in Figure 7. In order to prove the effectiveness of the pMyc/pMax NS strategy, the authors need to show the molecular biological data (e.g. signal pathway analysis related to cell death induced by disruption of the Myc:Max heterodimerization or their binding to the CME and non-CME in cells, etc.).

Round 3

Reviewer 2 Report

After revision, it is thought that this manuscript can be accepted for publication.